# Problem-solving interventions and depression among adolescents and young adults: A systematic review of the effectiveness of problem-solving interventions in preventing or treating depression

**Kristina Metz**[1]\*, **Jane Lewis**[2], **Jade Mitchell**[2], **Sangita Chakraborty**[2], **Bryce D. McLeod**[3], **Ludvig Bjørndal**[2], **Robyn Mildon**[4], **Aron Shlonsky**[5]

**1** Bloomberg School of Public Health, Johns Hopkins University, Baltimore, MD, United States of America, **2** Centre for Evidence and Implementation, London, United Kingdom, **3** Department of Psychology, Virginia Commonwealth University, Richmond, VA, United States of America, **4** Centre for Evidence and Implementation, Melbourne, Victoria, Australia, **5** Department of Social Work, Monash University, Melbourne, Victoria, Australia

\* Kmetz7@jh.edu

## Abstract

Problem-solving (PS) has been identified as a therapeutic technique found in multiple evidence-based treatments for depression. To further understand for whom and how this intervention works, we undertook a systematic review of the evidence for PS's effectiveness in preventing and treating depression among adolescents and young adults. We searched electronic databases (*PsycINFO*, *Medline*, *and Cochrane Library*) for studies published between 2000 and 2022. Studies meeting the following criteria were included: (a) the intervention was described by authors as a PS intervention or including PS; (b) the intervention was used to treat or prevent depression; (c) mean or median age between 13–25 years; (d) at least one depression outcome was reported. Risk of bias of included studies was assessed using the Cochrane Risk of Bias 2.0 tool. A narrative synthesis was undertaken given the high level of heterogeneity in study variables. Twenty-five out of 874 studies met inclusion criteria. The interventions studied were heterogeneous in population, intervention, modality, comparison condition, study design, and outcome. Twelve studies focused purely on PS; 13 used PS as part of a more comprehensive intervention. Eleven studies found positive effects in reducing depressive symptoms and two in reducing suicidality. There was little evidence that the intervention impacted PS skills or that PS skills acted as a mediator or moderator of effects on depression. There is mixed evidence about the effectiveness of PS as a prevention and treatment of depression among AYA. Our findings indicate that pure PS interventions to treat clinical depression have the strongest evidence, while pure PS interventions used to prevent or treat sub-clinical depression and PS as part of a more comprehensive intervention show mixed results. Possible explanations for limited effectiveness are discussed, including missing outcome

**Data Availability Statement:** All relevant methods and data are within the paper and its Supporting Information files.

**Funding:** This work was commissioned by Wellcome Trust and was conducted independently by the evaluators (all named authors). No grant number is available. Wellcome Trust had no role in study design, data collection and analysis, decision to publish or preparation of the manuscript. The authors declare no financial or other competing interests, including their relationship and ongoing work with Wellcome Trust. This does not alter our adherence to PLOS ONE policies on sharing data and materials.

**Competing interests:** The authors have declared that no competing interests exist.

bias, variability in quality, dosage, and fidelity monitoring; small sample sizes and short follow-up periods.

## Introduction

Depression among adolescents and young adults (AYA) is a serious, widespread problem. A striking increase in depressive symptoms is seen in early adolescence [1], with rates of depression being estimated to almost double between the age of 13 (8.4%) and 18 (15.4%) [2]. Research also suggests that the mean age of onset for depressive disorders is decreasing, and the prevalence is increasing for AYA. Psychosocial interventions, such as cognitive-behavioural therapy (CBT) and interpersonal therapy (IPT), have shown small to moderate effects in preventing and treating depression [3–6]. However, room for improvement remains. Up to half of youth with depression do not receive treatment [7]. When youth receive treatment, studies indicate that about half of youth will not show measurable symptom reduction across 30 weeks of routine clinical care for depression [8]. One strategy to improve the accessibility and effectiveness of mental health interventions is to move away from an emphasis on Evidence- Based Treatments (EBTs; e.g., CBT) to a focus on discrete treatment techniques that demonstrate positive effects across multiple studies that meet certain methodological standards (i.e., common elements; 9). Identifying common elements allows for the removal of redundant and less effective treatment content, reducing treatment costs, expanding available service provision and enhancing scability. Furthermore, introducing the most effective elements of treatment early may improve client retention and outcomes [9–13].

A potential common element for depression intervention is problem-solving (PS). PS refers to how an individual identifies and applies solutions to everyday problems. D'Zurilla and colleagues [14–17] conceptualize effective PS skills to include a constructive attitude towards problems (i.e., a positive problem-solving orientation) and the ability to approach problems systematically and rationally (i.e., a rational PS style). Whereas maladaptive patterns, such as negative problem orientation and passively or impulsively addressing problems, are ineffective PS skills that may lead to depressive symptoms [14–17]. Problem Solving Therapy (PST), designed by D'Zurilla and colleagues, is a therapeutic approach developed to decrease mental health problems by improving PS skills [18]. PST focuses on four core skills to promote adaptive problem solving, including: (1) defining the problem; (2) brainstorming possible solutions; (3) appraising solutions and selecting the best one; and (4) implementing the chosen solution and assessing the outcome [14–17]. PS is also a component in other manualized approaches, such as CBT and Dialectical Behavioural Therapy (DBT), as well as imbedded into other wider generalized mental health programming [19, 20]. A meta-analysis of over 30 studies found PST, or PS skills alone, to be as effective as CBT and IPT and more effective than control conditions [21–23]. Thus, justifying its identification as a common element in multiple prevention [19, 24] and treatment [21, 25] programs for adult depression [9, 26–28].

PS has been applied to youth and young adults; however, no manuals specific to the AYA population are available. Empirical studies suggest maladaptive PS skills are associated with depressive symptoms in AYA [5, 17–23]. Furthermore, PS intervention can be brief [29], delivered by trained or lay counsellors [30, 31], and provided in various contexts (e.g., primary care, schools [23]). Given PS's versatility and effectiveness, PS could be an ideal common element in treating AYA depression; however, to our knowledge, no reviews or meta-analyses on PS's effectiveness with AYA specific populations exist. This review aimed to examine the effectiveness of PS as a common element in the prevention and treatment of depression for AYA

 

within real-world settings, as well as to ascertain the variables that may influence and impact PS intervention effects.

## Methods

### Identification and selection of studies

Searches were conducted using *PsycInfo*, *Medline*, *and Cochrane Library* with the following search terms: "problem-solving", "adolescent", "youth", and" depression,*"* along with filters limiting results to controlled studies looking at effectiveness or exploring mechanisms of effectiveness. Synonyms and derivatives were employed to expand the search. We searched grey literature using *Greylit.org* and *Opengrey.eu*, contacted experts in the field and authors of protocols, and searched the reference lists of all included studies. The search was undertaken on 4[th] June 2020 and updated on 11[th] June 2022.

Studies meeting the following criteria were included: (a) the intervention was described by authors as a PS intervention or including PS; (b) the intervention was used to treat or prevent depression; (c) mean or median age between 13–25 years; and (d) at least one depression outcome was reported. Literature in electronic format published post 2000 was deemed eligible, given the greater relevance of more recent usage of PS in real-world settings. There was no exclusion for gender, ethnicity, or country setting; only English language texts were included. Randomized controlled trials (RCTs), quasi-experimental designs (QEDs), systematic reviews/ meta-analyses, pilots, or other studies with clearly defined comparison conditions (no treatment, treatment as usual (TAU), or a comparator treatment) were included. We excluded studies of CBT, IPT, Acceptance and Commitment Therapy (ACT), Dialectical Behaviour Therapy (DBT), and modified forms of these treatments. These treatments include PS and have been shown to demonstrate small to medium effects on depression [13, 14, 32], but the unique contribution of PS cannot be disentangled. The protocol for this review was not registered; however, all data collection forms, extraction, coding and analyses used in the review are available upon inquiry from the first author.

### Study selection

All citations were entered into Endnote and uploaded to Covidence for screening and review against the inclusion/exclusion criteria. Reviewers with high inter-rater reliability (98%) independently screened the titles and abstracts. Two reviewers then independently screened full text of articles that met criteria. Duplicates, irrelevant studies, and studies that did not meet the criteria were removed, and the reason for exclusion was recorded (see S1 File for a list of excluded studies). Discrepancies were resolved by discussion with the team leads.

### Data extraction

Two reviewers independently extracted data that included: (i) study characteristics (author, publication year, location, design, study aim), (ii) population (age, gender, race/ethnicity, education, family income, depression status), (iii) setting, (iv) intervention description (therapeutic or preventative, whether PS was provided alone or as part of a more comprehensive intervention, duration, delivery mode), (v) treatment outcomes (measures used and reported outcomes for depression, suicidality, and PS), and (vi) fidelity/implementation outcomes. For treatment outcomes, we included the original statistical analyses and/or values needed to calculate an effect size, as reported by the authors. If a variable was not included in the study publication, we extracted the information available and made note of missing data and subsequent limitations to the analyses.

 

RCTs were assessed for quality (i.e., confidence in the study's findings) using the Cochrane Risk of Bias 2.0 tool [33] which includes assessment of the potential risk of bias relating to the process of randomisation; deviations from the intended intervention(s); missing data; outcome measurement and reported results. Risk of bias pertaining to each domain is estimated using an algorithm, grouped as: Low risk; Some concerns; or High risk. Two reviewers independently assessed the quality of included studies, and discrepancies were resolved by consensus.

We planned to conduct one or more meta-analyses if the studies were sufficiently similar. Data were entered into a summary of findings table as a first step in determining the theoretical and practical similarity of the population, intervention, comparison condition, outcome, and study design. If there were sufficiently similar studies, a meta-analysis would be conducted according to guidelines contained in the Cochrane Collaboration Handbook of Systematic Reviews, including tests of heterogeneity and use of random effects models where necessary.

## Results

The two searches yielded a total number of 874 records (after the removal of duplicates). After title and abstract screening, 184 full-text papers were considered for inclusion, of which 25 studies met the eligibility criteria and were included in the systematic review (Fig 1). Unfortunately, substantial differences (both theoretical and practical) precluded any relevant meta-analyses, and we were limited to a narrative synthesis.

### Risk of bias assessment

Risk of bias assessments were conducted on the 23 RCTs (Fig 2; assessments by study presented in S1 Table). Risk of bias concerns were moderate, and a fair degree of confidence in

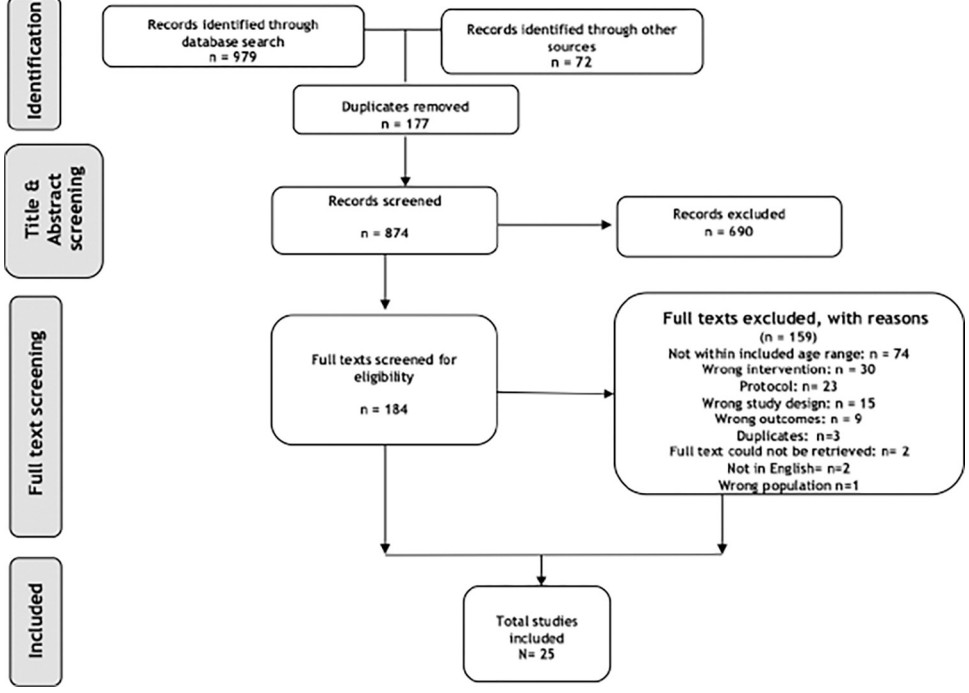

**Fig 1. PRISMA flow chart of the study selection process.**

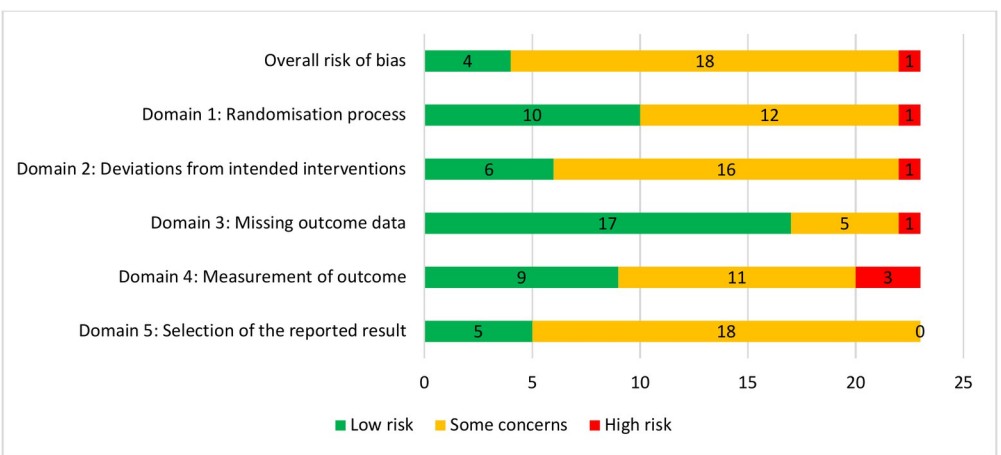

**Fig 2. Risk of bias summaries.**

the validity of study findings is warranted. Most studies (81%) were assessed as 'some concerns' (N = 18), four studies were 'low risk', and one 'high risk'. The most frequent areas of concern were the selection of the reported result (n = 18, mostly due to inadequate reporting of a priori analytic plans); deviations from the intended intervention (N = 17, mostly related to insufficient information about intention-to-treat analyses); and randomisation process (N = 13).

## Study designs and characteristics

**Study design.**   Across the 25 studies, 23 were RCTs; two were QEDs. Nine had TAU or wait-list control (WLC) comparator groups, and 16 used active control groups (e.g., alternative treatment). Eleven studies described fidelity measures. The sample size ranged from 26 to 686 and was under 63 in nine studies.

**Selected intervention.**   Twenty interventions were described across the 25 studies (Table 1). Ten interventions focused purely on PS. Of these 10 interventions: three were adaptations of models proposed by D'Zurilla and Nezu [20, 34] and D'Zurilla and Goldfried [18], two were based on Mynors-Wallis's [35] Problem-Solving Therapy (PST) guide, one was a problem-orientation video intervention adapted from D'Zurilla and Nezu [34], one was an online intervention adapted from Method of Levels therapy, and three did not specify a model. Ten interventions used PS as part of a larger, more comprehensive intervention (e.g., PS as a portion of cognitive therapy). The utilization and dose of PS steps included in these interventions were unclear. Ten interventions were primary prevention interventions–one of these was universal prevention, five were indicated prevention, and four were selective prevention. Ten interventions were secondary prevention interventions. Nine interventions were described as having been developed or adapted for young people.

**Intervention delivery.**   Of the 20 interventions, eight were delivered individually, eight were group-based, two were family-based, one was mixed, and in one, the format of delivery was unclear. Seventeen were delivered face-to-face and three online. Dosage ranged from a single session to 21, 50-minute sessions (12 weekly sessions, then 6 biweekly sessions); the most common session formar was once weekly for six weeks (N = 5).

**Intervention setting and participants.**   Seventeen studies were conducted in high-income countries (UK, US, Australia, Netherlands, South Korea), four in upper-middle income (Brazil, South Africa, Turkey), and four in low- and middle-income countries (Zimbabwe, Nigeria,

Table 1. Study characteristics.

| No. | STUDY DETAILS | INTERVENTION PURPOSE | INTERVENTION CHARACTERISTICS | STUDY DESIGN | CONTROL TREATMENT | STUDY POPULATION | OUTCOMES RELATIVE TO CONTROL AND MEASURE | COCHRANE RISK OF BIAS ASSESSMENT |
|---|---|---|---|---|---|---|---|---|
| PROBLEM SOLVING AS STANDALONE TREATMENT | | | | | | | | |
| 1 | Bird et al., 2018 UK | Selective prevention Problem-related distress | MYLO: Online, individual problem-solving program Self-delivered Duration participants' choice (minimum 15 minutes) Based on PCT principles | RCT N = 213 | ELIZA text-based programme emulating Rogerian psychotherapy | University students aged 16–70 years Mean age: 22.08 years No inclusion criteria for depression | No significant change in depression on the DASS-21. Main effect of group: ($F$ (1,157) = .16, n.s.). Interaction effect between time and group ($F$ (2,314) = .39, n.s.) No significant change in problem-related distress (study developed rating), time x group interaction: $F$ (2,338) = 1.32, $p$ = .27 No significant change in participants' ratings of problem resolution (study-developed rating). Main effect: ($F$ (1,60) = 2.49, n.s.) | Some concerns |
| 2 | Chibanda et al., 2014 Zimbabwe | Secondary prevention Postnatal depression | PST: Group, face-to-face PST intervention Delivered by trainer peer counselors 2, 60-minute sessions per week for 6 weeks Based on Mynors-Wallis (2005) | RCT N = 58 | Amitriptyline (antidepressant) and peer education | Women attending primary care clinics over the age of 18 Mean age: 25 years Met criteria for post-partum depression (DSM-1V) | Significant reduction in postnatal depression (EPDS) post intervention. PST group (M = 8.22, SD = 3.6), pharmacotherapy group (M = 10.7, SD = 2.7), $p$ = 0.0097. | Some concerns |
| 3 | Eskin et al., 2008 Turkey | Secondary prevention Depression | PST: Individual, face-to-face PST intervention Delivered by graduate clinical psychology students 6 weekly sessions Based on D'Zurilla and Goldfried (1971) and D'Zurilla and Nezu (1999) | RCT N = 46 | WLC | High school and university students Mean age: 19.1 years Diagnosed with MDD (SCIV) | Significant reduction in depression at the end of treatment on the BDI (ANOVA $F$(1, 42) = 10.3, $p$ < 0.01; adjusted effect size 1.6); HDRS ($F$(1, 42) = 37.7, $p$ < 0.0001; adjusted effect size 2.2) and suicide potential (SPS–$F$ (1, 42) = 7.3, $p$ < 0.05; adjusted effect size 0.21). No significant improvement in problem-solving skills on the PSI. Time x interaction effect: $F$(1, 42) = 2.2, $p$ > 0.05. Main effect for time: $F$(1, 42) = 6.4, $p$ < .05, with post-treatment scores being lower than pre-treatment. Follow-up PSI scores were significantly lower than pre-treatment ($Z$ = 3.7, $p$ < 0.0001) and post-treatment PSI scores ($Z$ = 2.0, $p$ < 0.05). Significant post-treatment depression recovery, BDI: 77% of PST participants and 15.8% of control participants achieved full or partial recovery; $x^2$ = 19.3, d.f. = 2, $p$ < 0.0001; HDRS: 96.3% of PST participants and 21.1% of control participants achieved full or partial recovery, $x^2$ = 31.1, d.f. = 2, $p$ < 0.0001. Follow-up BDI scores were statistically significantly lower than pre-treatment BDI scores ($Z$ = 4.1, $p$ < 0.0001), but similar to post-treatment BDI scores ($Z$ = 1.6, $p$ > 0.05). Follow-up HDRS scores were statistically significantly lower than pre-treatment HDRS scores ($Z$ = 4.1, $p$ < 0.0001), but similar to post-treatment HDRS scores ($Z$ = 0.1, $p$ > 0.05). | Some concerns |

*(Continued)*

**Table 1.** (Continued)

| | | | | | | |
|---|---|---|---|---|---|---|
| 4 | Fitzpatrick et al., 2005 US | Secondary prevention Suicidal ideation | Problem-orientation intervention: video focusing on PS and coping styles Self-delivered 35-minute video, one session with 2 modules to be completed Based on D'Zurilla and Nezu's (1999) Problem-Solving Therapy manual | RCT N = 110 | Single session video covering health issues including diet exercise and sleep | University students aged 18–24 years Mean age: 19.02 years With suicidal ideation ($\geq$ 6 BSS or endorsing active ideation) | Both suicidal ideation–BSS (Z = 2.17) and depression -BDI (Z = 2.72) had significant decreases pre- and post-treatment. However, these changes diminished over time. At follow-up, BSS differences were no longer significant and BDI differences were significant but small (ETA$^2$ = 010). No significant change in problem-solving skills or orientation SPSI-R. | Some concerns |
| 5 | Gaffney et al., 2014 UK | Selective prevention Problem-related distress | MYLO: Online, individual PS program Self-delivered Duration participants' choice (average time 19.23 minutes) Based on PCT principles | Pilot RCT N = 48 | ELIZA text-based programme emulating Rogerian psychotherapist | University students aged 18–32 years Mean age: 21.4 years No inclusion criteria for depression | No significant change in depression (DASS-21), time x group interaction: F (1.24, 49.57) = .50, p = .52 No significant change in problem-solving (study-developed rating), time x group interaction: F (1, 40) = 3.62, p = .06. No significant change in problem-related distress (study-developed rating), time x group interaction: F (2, 80) = 1.00, p = .37 | Some concerns |
| 6 | Hoek et al., 2012 Netherlands | Indicated prevention Depression | PST: Individual, online PST intervention Delivered by mental health care professionals 5 weekly sessions | RCT N = 45 | WLC | Recruitment through community and via parents treated for depression and anxiety aged 12–21 years Mean age: 16.07 years Self-report of mild or moderate depression and anxiety. Excluded severe depression (>40 on CES-D). | No significant change in depression (CES-D) after 4 months, group x time interaction: B = 0.54, SE = 1.14, p = 0.637. Recovery from clinical depression (CES-D) effects were not significantly different between intervention and waiting list groups.) | Some concerns |
| 7 | Houston et al., 2017 US | Universal prevention Resilience | Resilience and Coping Intervention: Group-based, face-to-face intervention to identify thoughts, feelings and coping strategies using PS techniques. Delivered by trained social workers 3, 45-minute weekly sessions | RCT N = 129 | TAU | University students aged 18–23 years Mean age: Not stated No inclusion criteria for depression | Significant reduction in depression (CES-D): F(1, 117) D = 5.36, p = .02; small effect size: Cohen's $f^2$ = 0.05. | Some concerns |

(Continued)

**Table 1.** (Continued)

| | | | | | | |
|---|---|---|---|---|---|---|
| 8 | Malik et al., 2021 India | Indicated prevention Common adolescent mental health problems | PS intervention: Individual face-to-face Delivered by college graduate counsellors with no formal training in psychological treatments 4–5, 30-minute sessions delivered over 2–3 weeks | RCT N = 251 | PS booklets without counsellor treatment | High school students aged 12–20 years Mean age: 15.61 years Elevated mental health symptoms and distress/functional impairment ($\geq$ 19 for boys and 20 for girls on SDQ Total Difficulties scale, $\geq$ 2 SDQ Impact Scale, > 1 month on SDQ chronicity index) | Significant reduction in psychosocial problems (YTP) at 12 months: adjusted mean difference = −0.75, 95% CI = [−1.47, −0.03], $p = 0.04$. Significant reduction in mental health symptoms (SDQ Total Difficulties Score) at 12 months: adjusted mean difference = −1.73, 95% CI = [−3.47, 0.02], $p = 0.05$. Significant intervention effect on both SDQ Total Difficulties and YTP scores over 12 months (SDQ Total Difficulties: adjusted mean difference = −1.23, 95% CI = [−2.37, −0.09]; d = 0.21, 95% CI = [0.05, 0.36]; $p = 0.03$; YTP: adjusted mean difference = −0.98; 95% CI = [−1.51, −0.45]; d = 0.34, 95% CI = [0.19, 0.50]; $p < 0.001$. Significant intervention effect on secondary outcomes including internalising symptoms (SDQ internalising symptoms subscale): adjusted mean difference = −0.76, 95% CI = [−1.42, −0.10]; d = 0.22, 95% CI = [0.06, 0.37]; $p = 0.03$; impairment (SDQ impact score): adjusted mean difference = −0.51, 95% CI = [−0.93, −0.09]; d = 0.21, 95% CI = [0.06, 0.36]; $p = 0.02$); and perceived stress (PSS-4): adjusted mean difference = −0.54 95% CI = [−1.00, −0.08]; d = 0.21, 95% CI = [0.06, 0.36]; $p = 0.02$) over 12 months. No significant effect on wellbeing (SWEMWBS): adjusted mean difference = 1.16, 95% CI = [−0.07, 2.38]; d = 0.19, 95% CI = [0.04, 0.34]; $p = 0.06$; externalising symptoms (SDQ externalising symptoms subscale): adjusted mean difference = −0.47, 95% CI = [−1.09, 0.14]; d = 0.14, 95% CI = [0.01, 0.30]; $p = 0.13$; or remission adjusted mean difference = 1.47, 95% CI = [0.73, 2.96]; $p = 0.28$ over 12 months. | Low risk of bias |

*(Continued)*

**Table 1.** (Continued)

| | | | | | | |
|---|---|---|---|---|---|---|
| 9 | Michelson et al., 2020 India | Indicated prevention Common adolescent mental health problems | PS intervention: Individual face-to-face Delivered by college graduate counsellors with no formal training in psychological treatments 4–5, 30-minute sessions delivered over 2–3 weeks | RCT N = 251 | PS booklets without counsellor treatment | High school students aged 12–20 years Mean age: 15.61 years Elevated mental health symptoms ($\geq$ 19 for boys and 20 for girls on SDQ Total Difficulties scale, $\geq$ 2 SDQ Impact Scale, > 1 month on SDQ chronicity index) | Significant reduction in psychosocial problems (YTP) at 6 weeks (adjusted mean difference = −1·01, 95% CI [−1·63, −0·38]; adjusted effect size = 0.36, 95% CI [0·11, 0·61], $p$ = 0·0015), and at 12 weeks (adjusted mean difference = −1.03, 95% CI [−1·60, −0·47]; adjusted effect size = 0.35, 95% CI [0.18, 0·54]; $p$ = 0·0004). No significant change in mental health symptoms (SDQ Total Difficulties score) at 6 weeks (adjusted mean difference = −0·86, 95% CI [−2·14, 0.41]; adjusted effect size = 0·16, 95% CI [0·09, 0.41]; $p$ = 0·18) or 12 weeks (adjusted mean difference = −1.12, 95% CI [−2.33, 0.10]; adjusted effect size = 0.20, 95% CI [0.02, 0.37], $p$ = 0.072. No significant change in internalising symptoms (SDQ Internalising symptoms subscale) at 12 weeks (adjusted mean difference = −0.61, 95% CI [−1.32, 0.09]; adjusted effect size = 0.18, 95% CI [0.002, 0.36], $p$ = 0.089. | Low risk of bias |
| 10 | Parker et al., 2016 Australia | Indicated prevention Depression | PST: Face-to-face; Not specified whether group or individual Delivered by research psychologists 6 weekly sessions Based on Mynors-Wallis (2005) | RCT N = 176 | Control treatment based on general counselling principles informed by NICE guidelines for mild to moderate depression | Young people recruited from youth mental health services aged 15–25 years Mean age: 17.6 years Elevated symptoms not specific to depression indicating a mild disorder (K10 score $\geq$ 20) | No significant change in depression on the BDI-II (difference in change between interventions = 0.13, 95% CI [−3.10, 3.36], $p$ = 0.935) or on the MADRS (difference in change between interventions = 0.29, 95% CI [−2.66, 3.24], $p$ = 0.847. | Some concerns |
| 11 | Tezel & Gözüm, 2006 Turkey | Indicated prevention Postnatal depression | PST: Individual, face-to-face Delivered by nurse researchers 6, 30–50-minute weekly sessions Based on D'Zurilla and Goldfield (1971) | QED N = 62 | Nursing intervention | Mothers vising postnatal care service Mean age: 24.6 years (care group); 25.4 years (training group) At risk of post-partum depression (>11 EPDS) but without major depressive symptoms | Significant reduction pre-test to post-test in depression (BDI) for PST ($t$ = 5.462, $p$ < 0.05) and the nursing intervention ($t$ = 10.062, $p$ < 0.05). However, the nursing intervention was significantly more effective than PST at reducing depressive symptoms ($t$ = 4.529, $p$ < 0.05). | NA |

*(Continued)*

**Table 1.** (Continued)

| No. | STUDY DETAILS | INTERVENTION PURPOSE | INTERVENTION CHARACTERISTICS | STUDY DESIGN | CONTROL TREATMENT | STUDY POPULATION | OUTCOMES RELATIVE TO CONTROLE AND MEASURE | |
|---|---|---|---|---|---|---|---|---|
| 12 | Xavier et al., 2019 Brazil | Secondary prevention Suicidal behaviour | PST: Group, face-to-face Delivered by experienced psychologist 5, 120-minute weekly sessions Based on D'Zurilla and Nezu (2007) and Vazquez et al. 2015) [70] | RCT N = 100 | TAU | Poorly performing students aged 15–19 years recruited from 3 public schools Mean age: 17.2 years Met criteria for depression (≥ 16 CES-D) and high risk of suicide (score total ≥ 45 or critical item score ≥3 on ISO-30) but not major depression | Significant intervention effect for depression (CES-D) across time points: $F$ $(3.58, 351.24) = 140.81, p < .001, η2 = 0.59$). Post-treatment: $t = 28.00, d = 5.60$, 95% CI [4.57, -6.60]. 6-month follow-up: $t = 22.65, t = 4.53$, 9%% CI [3.72, -3.26]. Significant intervention effect for suicidal orientation (ISO-30): $F(3.39, 312.51) = 104.75, p < .001, η2 = 0.52$. Significantly more participants no longer at risk of suicide in the PST group (96%) compared to the control group (0%) at post-test ($x^2(1) = 92.3, p < 0.001$) and at 6-month follow-up ($x^2(1) = 92.3, p <0.001$). Post-treatment: $t = 30.29, d = 6.05$, 95% CI [5.11–6.99]. 6-month follow-up: $t = 14.08, d = 2.82$, 95% CI [2.21, -3.41]. No significant difference in suicide plans or attempts ($p = .495$). global problem-solving skills and in functional problem-solving skills (SPSR-I) mediated the relationship between the experimental condition and the pre-/posttreatment change in suicidal orientation, with significant effects of mediation of 20.46, 95% CI [23.32, 57.35] and 13.99, 95% CI [33.18–57.96], respectively. These explained 34.5% and 23.6% of the total effect of the intervention on the change in suicidal orientation. | Low risk of bias |
| **PROBLEM-SOLVING AS PART OF WIDER INTERVENTION** | | | | | | | | |
| 13 | Brugha et al., 2000 UK | Indicated prevention Postnatal depression | Preparing for Parenthood: Group, face-to-face cognitive, PS, and social support intervention Delivered by nurses and occupational therapists 6, 120-minute weekly sessions Based on an international, collaborative review of the social support intervention literature (Brugha, 1995). [62] | RCT N = 292 | TAU | Mothers attending antenatal clinics aged 16-38 years Median age: 19 years Increased risk of post-natal depression (1+ items on modified GHQ) | No significant change in postnatal depression (EPDS: OR = 0.82, 95% CI [0.39, 1.75], $p = 0.61$); GHQ-D: OR = 1.22, 95% CI [0.63, 2.39], $p = 0.55$); SCAN ICD-10: OR = 0.48, 95% CI [0.12, 1.99], $p = 0.30$). Intervention group significantly more likely to adopt an avoidant problem-solving style (OR = 2.23, 95% CI [1.23, 4.06], $p = .009$). No significant group differences in confidence in ability to solve problems (OR = 0.69, 95% CI [0.25, 1.90], $p = 0.48$) or belief in personal control when solving problems (OR = 1.13, 95% CI [0.64, 2.00], $p = 0.67$). | Some concerns |

*(Continued)*

**Table 1.** (Continued)

| | | | | | | |
|---|---|---|---|---|---|---|
| 14 | Dietz et al., 2014 US | Secondary prevention Depression | SBFT aimed to treat family dysfunction and teach PS skills to families. Delivered by trained therapists. Phase 1 involved 12–16 weekly sessions, phase 2 involved 2–4 booster sessions | RCT N = 63 | CBT or NST | Patients recruited from 2 mental health clinics aged 13–18 years. Mean age: 15.6 years. Met DSM criteria for MDD ($\geq 13$ BDI) | This report focused on whether the PS components of CBT and SFBT mediated the effectiveness of these interventions for remission of major depressive disorder. PS mediated the association between CBT, but not SFBT, and remission from depression such that there was no significant association between SBFT and remission status (K-SADS-P; Wald $z = 0.00$, $p = 0.99$) and there was a significant association between CBT and remission status (K-SADS-P; Wald $z = 4.64$, $p = 0.03$). CBT (B = 0.41, CI [.29, 1.67], $t = 2.85$, $p = 0.0006$) and SFBT (B = 0.30, CI [0.2, 1.47], $t = 2.07$, $p = 0.04$) were both associated with increased PS (video rating). | Some concerns |
| 15 | Gureje et al., 2019 Nigeria | Secondary prevention Perinatal depression | Individual, face-to-face PS intervention. Delivered by primary maternal care providers. 8, 30–45-minute initial weekly sessions, followed by 4, 30–45-minute fortnightly sessions, third stage with option of pharmacotherapy/specialist referral for patients with higher EPDS scores. Adapted from PST-PC. | RCT N = 686 | Enhanced care as usual including psycho-education and social support | Mothers attending childcare clinics aged 16–45 years. Mean age: 24.7 years. Met criteria for major depression ($\geq 12$ EPDS) | Significant reduction in depression symptoms (EPDS) at 6 months: between group adjusted mean difference over four follow-up time points: -0.8, 95% CI [-1.3, 0.2], p = 0.007. No significant difference in remission rate (EPDS): adjusted risk difference = 4%, 95% CI [-4.1%, 12.0%]. Significant increase in remission rate for subgroup of women with more severe baseline depression compared to control (OR = 2.29, 95% CI [1.01, 5.20,] p = 0.047). | Some concerns |
| 16 | Haeffel et al., 2017 US | Selective prevention Depression | Social Problem-Solving Therapy: Group, face-to-face intervention designed to increase social PS and social skills. Delivered by trained correctional officers. 10, 60-minute sessions. Based on the Viewpoints manual (Guerra, Moore, & Slaby, 1995 [66]; Guerra & Slaby, 1990 [65]; Guerra & Williams, 2012).[67] | RCT N = 296 | TAU–psychosocial support | Juvenile detainees in state-run detention centres aged 11–16 years. Mean age: 14.97 years. No inclusion criteria for depression | No significant change in depression (CDI) compared to control: $F(1, 139) = 0.02$, $p = 0.89$, $\eta^2 p < 0.01$. Significant reduction in depression (CDI) for sub-group with higher intelligence and significant increase in depression for participants with lower intelligence: $B = -0.34$, $t = -2.26$, $p = 0.03$, partial correlation = -.19, change in $R^2 = .02$. Simple slope gradient for those with higher and lower levels of intelligence was significantly different depending on intervention type: $t = -2.11$, $p = 0.03$, partial correlation = -.27; effect size in the medium range. | High risk of bias |
| 17 | Hallford & Mellor, 2016 Australia | Secondary prevention Depression | Cognitive Reminiscence Therapy: Individual, face-to-face cognitive therapy that included brief PST. Delivered by registered provisional psychologist. 6 weekly sessions. Based on the protocol by Watt and Cappeliez (2000) [71] | RCT N = 26 | Brief evidence-based treatment | Young people recruited from a community youth mental health service aged 12–25 years. Mean age: 20.8 years. At least moderate depression (score $\geq 7$ on DASS-21) | No significant reduction in depression (DASS-21: $F(1, 24) = 1.9$, p = 0.146) | Some concerns |

*(Continued)*

**Table 1.** (Continued)

| | Study | Prevention type / focus | Intervention | Design / N | Comparison | Population | Results | Risk of bias |
|---|---|---|---|---|---|---|---|---|
| 18 | Hood et al., 2018 US | Selective prevention Diabetes distress | Penn Resilience Program Type 1 Diabetes: Group, face-to-face resilience enhancing intervention with a focus on diabetes management. Teaches cognitive-behavioural, social, and PS skills. Delivered by masters-level clinicians 9, 90–120 minute bi-weekly sessions Adapted from the University of Pennsylvania Penn Resilience Program (Gillham et al., 2006) | RCT N = 264 | Diabetes educational intervention | Patients from diabetes clinics aged 14–18 years Mean age: 15.74 years No inclusion criteria for depression—excluded with depression diagnosis or treatment | No significant reduction in depression (CDI). Symptoms remined stable over time across groups (slope intercept $p$ values > 0.05) and no significant group differences found (treatment-intercept and treatment-slope effect $p$ values > 0.05). No significant reduction in PS (SPSR-I) between groups ($p$>.05). | Some concerns |
| 19 | Kolko et al. (2000) | Secondary prevention Depression | SBFT aimed to treat family dysfunction and teach PS skills to families. Delivered by trained therapists Phase 1 involved 12–16 weekly sessions, phase 2 involved 2–4 booster sessions Based on Functional Family Therapy (Alexander & Parsons, 1982) [61] and the PS model developed by Robin and Foster (1989) [69] | RCT N = 107 | CBT or NST | Patients recruited from 2 mental health clinics aged 13–18 years Mean age: 15.6 years Met DSM criteria for MDD ($\geq$ 13 BDI) | No significant reduction in depression post treatment (BDI treatment x time interaction: $p$ < .08; DEP-13 treatment x time interaction: $p$ < .41) or after 24-months follow-up (BDI: $p$ < .62; DEP-13: $p$ < .92) | Some concerns |
| 20 | Makover et al, 2019 US | Secondary prevention Depression | High School Transition Program: Group and individual, face-to-face intervention designed to increase social and academic PS skills Delivered by trained mental health counsellors 12, 60 -minute group sessions followed by 4 individual booster sessions Based on CAST (Eggert et al., 2002) [63] | RCT N = 497 | Interview and clinical follow-up without active therapy | Middle and high school students in $8^{th}$ and $9^{th}$ grade Mean age: Not stated Met criteria for depression (score $\geq$ 15 on MFQ) | No significant reduction in depression (MFQ: $X_2(1) = 2.18, p = .08$) | Some concerns |
| 21 | Miklowitz et al, 2014 US | Secondary prevention Mood episodes (including BP and MDD) | Family-Focused Therapy: Face-to-face family-based sessions including psychoeducation, communication, and PS skills training. Delivered by trained therapists 21, 50-minute weekly/bi-weekly sessions | RCT N = 145 | Pharmacotherapy | Adolescents with a DSM-IV-TR diagnosis of bipolar I or II disorder aged 12–18 years Mean age: 15.6 years Symptoms of at least moderate severity (a score >17 on the K-SADS Mania Rating Scale or a score >16 on the Depression Rating Scale) | No significant group differences in time No significant group differences in time free of mood symptoms (depressive or manic symptoms), or percentage of weeks with mood symptoms or depressive symptoms. Family-focused therapy had a greater increase from year 1 to year 2 than enhanced care in the proportion of weeks without mania/hypomania symptoms (F = 4.02, df =1, 87, p = 0.048) Family-focused treatment showed greater improvements in mean Psychiatric Status Rating Scale scores for mania/hypomania across 3-month intervals than enhanced care (F = 1.98, df = 8, 742, p = 0.046) | Some concerns |

*(Continued)*

**Table 1.** (Continued)

| | | | | | | | Low risk of bias / NA / Some concerns |
|---|---|---|---|---|---|---|---|
| 22 | Miłkowitz et al., 2020 US | Secondary prevention Mood episodes (including BP and MDD) | Family-Focused Therapy: Face-to-face family-based sessions including psychoeducation, communication, and PS skills training Delivered by trained therapists 12, 60-minute weekly/bi-weekly sessions | RCT N = 127 | Enhanced care (EC) including family and individual psychoeducation | High risk youths aged 9–17 years and their parents Mean age: 13.2 years Met DSM-IV/DSM-5 criteria for BD or MDD First or second degree relative with lifetime history of BD-I or BD-II Moderate current mood symptoms (prior week YMRS ≥ 11 or 2-week CDRS-R > 29) | No significant difference in time to recovery. In the Family Focused Therapy (FFT group), 47 of 61 participants (77.0%) recovered in a median of 24 weeks (95% CI, 17–33 weeks) compared with 43 of 66 (65.2%) in the EC group in 23 weeks (95% CI, 17–29 weeks) (log-rank $\chi2 = 0.01$; $P = .93$; unadjusted hazard ratio [HR] for FFT vs EC, 1.02; 95% CI, 0.67–1.54). Among participants who recovered (N = 90) FFT participants experienced longer times with- out a new mood episode than EC participants (unadjusted, treatment of the treated log-rank $\chi2 = 5.44$; $P = .02$; HR, 0.55; 95% CI, 0.48–0.92). The estimated median time from randomization to a new mood episode was 73 weeks (95% CI, 55–82 weeks) in the intent-to- treat sample (n = 127), with a median of 81 weeks (95% CI, 56–123 weeks) for those in the FFT group and 63 weeks (95% CI, 44–78 weeks) for those in the EC group. Patients in the FFT group had longer intervals of wellness before new mood episodes than patients in the EC group ($\chi2 = 4.44$; $P = .03$; HR, 0.59; 95% CI, 0.35–0.97). Significantly longer intervals between recovery and next mood episode (A-LIFE and PSRs: $\chi2 = 5.44$; $P = .02$; hazard ratio, 0.55; 95% CI, 0.48–0.92;), and from randomisation to the next mood episode (A-LIFE and PSRs: $\chi2 = 4.44$; $P = .03$; hazard ratio, 0.59; 95% CI, 0.35–0.97). | Low risk of bias |
| 23 | Noh, 2018 South Korea | Selective prevention Build resilience and reduce impact of trauma | Resilience enhancement program: Group, face-to-face intervention designed to increase resilience with a component on PS Delivered by the author and a psychiatric nurse 2, 90-minute sessions per week for 4 weeks | QED N = 32 | TAU by youth shelters | Runaway youths from homeless shelters aged 12–21 years Mean age: 16.69 years No inclusion criteria for depression—excluded young people receiving psychiatric interventions | Significant reduction in depression (BDI-II) at post-test (beta = -5.33, $p = 0.036$) but not at one-month follow-up (beta = -4.48, $p = 0.120$). | NA |
| 24 | Psaros et al. 2022 South Africa | Secondary prevention Depression | PST: Individual, face-to-face PST plus LifeSteps adherence intervention Delivered by a trained lay counsellor 8 weekly sessions Based on PST (Bell & D'Zurilla, 2009; Nezu, Maguth Nezu & D'Zurilla 2013) [68] | RCT N = 23 | TAU | Pregnant women with HIV aged 18–45 years Median age: 24 years Met criteria for current major depressive episode (data from a structured clinical interview, self-report measure, and team consensus based on clinical impressions) | Significant reduction in depression (BDI-II) at post-test (beta = -5.33, $p = 0.036$) but not at one-month follow-up (beta = -4.48, $p = 0.120$). | Some concerns |

(Continued)

**Table 1.** (Continued)

| | | | | | |
|---|---|---|---|---|---|
| 25 | Weissberg-Benchell et al., 2020 US | Selective prevention Diabetes distress | Penn Resilience Program Type 1 Diabetes: Group, face-to-face resilience enhancing intervention with a focus on diabetes management. Teaches cognitive-behavioural, social and PS skills. Delivered by masters-level clinicians 9, 90-120-minute bi-weekly sessions Adapted from the University of Pennsylvania Penn Resilience Program (Gillham et al., 2006) [64] | RCT N = 264 | Diabetes educational intervention | Patients from diabetes clinics aged 14–18 years Mean age: 15.7 years No inclusion criteria for depression—excluded with depression diagnosis or treatment | Stable depressive symptoms (CDI) from 0 to 16 months (slope₁: $b = 0.41$, $SE = 0.28$, $p = .139$, $\beta = 0.16$; quadratic slope $b = 0.07$, $SE = 0.07$, $p = .269$, $\beta = 0.10$). Decline in depressive symptoms from 16 to 40 months (slope₁: $b = -0.17$, $SE = 0.07$, $p = .018$, $\beta = 0.20$). The effect size of change in depressive symptoms from 16 to 40 months was $d = 0.12$. Follow-up assessment of change in depressive symptoms from 16 to 40 months separated by intervention group indicated that there was a significant decline in depressive symptoms for the intervention participants, $b = -0.31$, $SE = 0.11$, $p = .005$, $\beta = -0.31$, but not for control participants, $b = -0.01$, $SE = 0.09$, $p = .936$, $\beta = -0.01$. No significant difference in depressive symptoms at 40 months, $b = -1.76$, $SE = 0.94$, $p = .060$, $\beta = -0.13$, $d = 0.23$. | Some concerns |

Notes: Psychiatric measures: A-LIFE = Adolescent Longitudinal Interval Follow-up Evaluation; BDI = Beck Depression Inventory; BDI-II = Beck Depression Inventory-II; BSS = Beck Suicide Scale; CDI = Children's Depression Inventory; CDRS-R = Children's Depression Rating Scale, Revised; CES-D = Centre for Epidemiological Studies Depression Scale; DASS-21 = Depression Anxiety Stress Scale-21; DEPI3 = 13 items from Schedule for Affective Disorders and Schizophrenia for School-Age Children; DSM-IV = Diagnostic and Statistical Manual of Mental Health Disorders, fourth edition; DSM-IV-TR = Diagnostic and Statistical Manual of Mental Health Disorders, fourth edition, text revision; DSM-5 = Diagnostic and Statistical Manual of Mental Health Disorders, fifth edition; EPDS = Edinburgh Postnatal Depression Scale; GHQ = General Health Questionnaire; GHQ-D = General Health Questionnaire Depression Scale; HDRS = Hamilton Depression Rating Scale; ICD-10 = International Classification of Diseases, 10th revision

ISO-30 = Inventory of Suicide Orientation; K10 = Kessler Psychological Distress Scale; K-SADS = Kiddie Schedule for Affective Disorders and Schizophrenia; MADRS = Montgomery–Åsberg Depression Rating Scale; MFQ = Mood and Feelings Questionnaire; PSRs = Psychiatric Status Ratings; PSS-4 = Perceived Stress Scale-4; SCAN = Schedules for Clinical Assessment in Neuropsychiatry; SCIV = Structured Clinical Interview Clinical Version for DSM-IV Axis 1; SDQ = Strengths and Difficulties Questionnaire; SPS = Suicide Probability Scale; SWEMWBS = Short Warwick–Edinburgh Mental Well-Being Scale; YMRS = Young Mania Rating Scale; YTP = Youth Top Problems

Problem-solving measures: PSI = Problem Solving Inventory; SPSI = Social Problem-Solving Inventory; SPSI-R = Social Problem-Solving Inventory-Revised

Other terms: CAST = Coping and Support Training; CBT = Cognitive Behavioural Therapy; MYLO = Manage Your Life Online; NICE = National Institute of Health and Care Excellence; NST = Nondirective Supportive Therapy; PCT = Perceptual Control Therapy; PS = Problem Solving; PST = Problem-Solving Therapy; PST-PC = Problem-Solving Therapy for Pediatric Care; QED = Quasi-Experimental Design; RCT = Randomized Controlled Trial; SBFT = Systematic-Behavioural Family Therapy; TAU = Treatment As Usual; WLC = Waitlist Control; BP = Bipolar Disorder; MDD = major depressive disorder

India). Four studies included participants younger than 13 and four older than 25. Nine studies were conducted on university or high school student populations and five on pregnant or post-partum mothers. The remaining 11 used populations from mental health clinics, the community, a diabetes clinic, juvenile detention, and a runaway shelter.

Sixteen studies included participants who met the criteria for a depressive, bipolar, or suicidal disorder (two of these excluded severe depression). Nine studies did not use depression symptoms in the inclusion criteria (one of these excluded depression). Several studies excluded other significant mental health conditions.

**Outcome measures.** Eight interventions targeted depression, four post/perinatal depression, two suicidal ideation, two resilience, one 'problem-related distress', one 'diabetes distress', one common adolescent mental health problem, and one mood episode. Those targeting post/perinatal depression used the Edinburgh Postnatal Depression Scale as the outcome measure. Of the others, six used the Beck Depression Inventory (I or II), two the Children's Depression Inventory, three the Depression Anxiety Stress Scale-21, three the Centre for Epidemiologic Studies Depression Scale, one the Short Mood and Feelings Questionnaire, one the Hamilton Depression Rating Scale, one the depression subscale on the Schedule for Affective Disorders and Schizophrenia for School-Age Children, one the Strengths and Difficulties Questionnaire, one the Youth Top Problems Score, one the Adolescent Longitudinal Interval Follow-up Evaluation and Psychiatric Status Ratings, one the Kiddie Schedule for Affective Disorders and Schizophrenia, and one the Mini International Neuropsychiatric Interview.

Only eight studies measured PS skills or orientation outcomes. Three used the Social Problem-Solving Inventory-Revised, one the Problem Solving Inventory, two measured the extent to which the nominated problem had been resolved, one observed PS in video-taped interactions, and one did not specify the measure.

## Outcomes

The mixed findings regarding the effectiveness of PS for depression may depend on the type of intervention: primary (universal, selective, or indicated), secondary or tertiary prevention. Universal prevention interventions target the general public or a population not determined by any specific criteria [36]. Selective prevention interventions target specific populations with an increased risk of developing a disorder. Indicated prevention interventions target high-risk individuals with sub-clinical symptoms of a disorder. Secondary prevention interventions include those that target individuals diagnosed with a disorder. Finally, tertiary prevention interventions refer to follow-up interventions designed to retain treatment effects. Outcomes are therefore grouped by intervention prevention type and outcome. Within these groupings, studies with a lower risk of bias (RCTs) are presented first. According to the World Health Organisation guidelines, interventions were defined as primary, secondary or tertiary prevention [36].

## Universal prevention interventions

One study reported on a universal prevention intervention targeting resilience and coping strategies in US university students. The Resilience and Coping Intervention, which includes PS as a primary component of the intervention, found a significant reduction in depression compared to TAU (RCT, $N = 129$, moderate risk of bias) [37].

## Selective prevention interventions

Six studies, including five RCTs and one QED, tested PS as a selective prevention intervention. Two studies investigated the impact of the Manage Your Life Online program, which includes PS as a primary component of the intervention, compared with an online programme emulating Rogerian psychotherapy for UK university students (RCT, $N$ = 213, moderate risk of bias [38]; RCT, $N$ = 48, moderate risk of bias [39]). Both studies found no differences in depression or problem-related distress between groups.

Similarly, two studies explored the effect of adapting the Penn Resilience Program, which includes PS as a component of a more comprehensive intervention for young people with diabetes in the US (RCT, $N$ = 264, moderate risk of bias) [40, 41]. The initial study showed a moderate reduction in diabetes distress but not depression at 4-, 8-, 12- and 16-months follow-up compared to a diabetes education intervention [40]. The follow-up study found a significant reduction in depressive symptoms compared to the active control from 16- to 40-months; however, this did not reach significance at 40-months [41].

Another study that was part of wider PS and social skills intervention among juveniles in state-run detention centres in the US found no impacts (RCT, $N$ = 296, high risk of bias) [42]. A QED ($N$ = 32) was used to test the effectiveness of a resilience enhancement and prevention intervention for runaway youth in South Korea [43]. There was a significant decrease in depression for the intervention group compared with the control group at post-test, but the difference was not sustained at one-month follow-up.

## Indicated prevention interventions

Six studies, including five RCTs and one QED, tested PS as an indicated prevention intervention. Four of the five RCTs tested PS as a primary component of the intervention. A PS intervention for common adolescent mental health problems in Indian high school students (RCT, $N$ = 251, low risk of bias) led to a significant reduction in psychosocial problems at 6- and 12 weeks; however, it did not have a significant impact on mental health symptoms or internalising symptoms compared to PS booklets without counsellor treatment at 6- and 12-weeks [31]. A follow-up study showed a significant reduction in overall psychosocial problems and mental health symptoms, including internalizing symptoms, over 12 months [44]. Still, these effects no longer reached significance in sensitivity analysis adjusting for missing data (RCT, $N$ = 251, low risk of bias). Furthermore, a 2x2 factorial RCT ($N$ = 176, moderate risk of bias) testing PST among youth mental health service users with a mild mental disorder in Australia found that the intervention was not superior to supportive counselling at 2-weeks post-treatment [30]. Similarly, an online PS intervention delivered to young people in the Netherlands to prevent depression (RCT, $N$ = 45, moderate risk of bias) found no significant difference between the intervention and WLC in depression level 4-months post-treatment [45].

One RCT tested PS approaches in a more comprehensive manualized programme for postnatal depression in the UK and found no significant differences in depression scores between intervention and TAU at 3-months post-partum (RCT, $N$ = 292, moderate risk of bias) [46].

A study in Turkey used a non-equivalent control group design (QED, $N$ = 62) to test a nursing intervention against a PS control intervention [47]. Both groups showed a reduction in depression, but the nursing care intervention demonstrated a larger decrease post-intervention than the PS control intervention.

## Secondary prevention interventions

Twelve studies, all RCTs, tested PS as a secondary prevention intervention. Four of the 12 RCTs tested PS as a primary component of the intervention. An intervention among women

in Zimbabwe (RCT, $N$ = 58, moderate risk of bias) found a larger decrease in the Edinburgh Postnatal Depression Scale score for the intervention group compared to control (who received the antidepressant amitriptyline and peer education) at 6-weeks post-treatment [48]. A problem-orientation intervention covering four PST steps and involving a single session video for US university students (RCT, $N$ = 110, moderate risk of bias), compared with a video covering other health issues, resulted in a moderate reduction in depression post-treatment; however, results were no longer significant at 2-weeks, and 1-month follow up [49].

Compared to WLC, a study of an intervention for depression and suicidal proneness among high school and university students in Turkey (RCT, $N$ = 46, moderate risk of bias) found large effect sizes on post-treatment depression scores for intervention participants post-treatment compared with WLC. At 12-month follow-up, these improvements were maintained compared to pre-test but not compared to post-treatment scores. Significant post-treatment depression recovery was also found in the PST group [12]. Compared to TAU, a small but high-quality (low-risk of bias) study focused on preventing suicidal risk among school students in Brazil (RCT, $N$ = 100, low risk of bias) found a significant, moderate reduction in depression symptoms for the treatment group post-intervention that was maintained at 1-, 3- and 6-month follow-up [50].

Seven of the 12 RCTs tested PS as a part of a more comprehensive intervention. Two interventions targeted mood episodes and were compared to active control. These US studies focused on Family-Focused Therapy as an intervention for mood episodes, which included sessions on PS [51, 52]. One of these found that Family-Focused Therapy for AYA with Bipolar Disorder (RCT, $N$ = 145, moderate risk of bias) had no significant impact on mood or depressive symptoms compared to pharmacotherapy. However, Family-Focused Therapy had a greater impact on the proportion of weeks without mania/hypomania and mania/hypomania symptoms than enhanced care [53]. Alternatively, while the other study (RCT, $N$ = 127, low risk of bias) found no significant impact on time to recovery, Family-Focused Therapy led to significantly longer intervals of wellness before new mood episodes, longer intervals between recovery and the next mood episode, and longer intervals of randomisation to the next mood episode in AYA with either Bipolar Disorder (BD) or Major Depressive Disorder (MDD), compared to family and individual psychoeducation [52].

Two US studies used a three-arm trial to compare Systemic-behavioural Family Therapy (SBFT) with elements of PS, to CBT and individual Non-directive Supportive therapy (NST) (RCT, $N$ = 107, moderate risk of bias) [53, 54]. One study looked at whether the PS elements of CBT and SFBT mediated the effectiveness of these interventions for the remission of MDD. It found that PS mediated the association between CBT, but not SFBT, and remission from depression. There was no significant association between SBFT and remission status, though there was a significant association between CBT and remission status [53]. The other study found no significant reduction in depression post-treatment or at 24-month follow-up for SBFT [54].

A PS intervention tested in maternal and child clinics in Nigeria RCT ($N$ = 686, moderate risk of bias) compared with enhanced TAU involving psychosocial and social support found no significant difference in the proportion of women who recovered from depression at 6-months post-partum [55]. However, there was a small difference in depression scores in favour of PS averaged across the 3-, 6-, 9-, and 12-month follow-up points. Cognitive Reminiscence Therapy, which involved recollection of past PS experiences and drew on PS techniques used for 12-25-year-olds in community mental health services in Australia (RCT, $N$ = 26, moderate risk of bias), did not reduce depression symptoms compared with a brief evidence-based treatment at 1- or 2-month follow-up [56]. Additionally, the High School Transition Program in the US (RCT, N = 497, moderate risk of bias) aimed to prevent depression, anxiety, and

school problems in youth transitioning to high school [57]. There was no reduction in the percentage of intervention students with clinical depression compared to the control group. Similarly, a small study focused on reducing depression symptoms, and nonadherence to antiretroviral therapy in pregnant women with HIV in South Africa (RCT, $N$ = 23, some concern) found a significant reduction in depression symptoms compared to TAU, with the results being maintained at the 3-month follow-up [58].

## Reduction in suicidality

Three studies measured a reduction in suicidality. A preventive treatment found a large reduction in suicidal orientation in the PS group compared to control post-treatment. In contrast, suicidal ideation scores were inconsistent at 1-,3- and 6- month follow-up, they maintained an overall lower score [50]. Furthermore, at post-test, significantly more participants in the PS group were no longer at risk of suicide. No significant differences were found in suicide plans or attempts. In a PST intervention, post-treatment suicide risk scores were lower than pre-treatment for the PST group but unchanged for the control group [12]. An online treatment found a moderate decline in ideation for the intervention group post-treatment compared to the control but was not sustained at a one-month follow-up [49].

## Mediators and moderators

Eight studies measured PS skills or effectiveness. In two studies, despite the interventions reducing depression, there was no improvement in PS abilities [12, 52]. One found that change in global and functional PS skills mediated the relationship between the intervention group and change in suicidal orientation, but this was not assessed for depression [50]. Three other studies found no change in depression symptoms, PS skills, or problem resolution [38–40]. Finally, CBT and SBFT led to significant increases in PS behaviour, and PS was associated with higher rates of remission across treatments but did not moderate the relationship between SBFT and remission status [53]. Another study found no changes in confidence in the ability to solve problems or belief in personal control when solving problems. Furthermore, the intervention group was more likely to adopt an avoidant PS style [46].

A high-intensity intervention for perinatal depression in Nigeria had no treatment effect on depression remission rates for the whole sample. Still, it was significantly effective for participants with more severe depression at baseline [55]. A PS intervention among juvenile detainees in the US effectively reduced depression for participants with higher levels of fluid intelligence, but symptoms increased for those with lower levels [42].The authors suggest that individuals with lower levels of fluid intelligence may have been less able to cope with exploring negative emotions and apply the skills learned.

## Discussion

This review has examined the evidence on the effectiveness of PS in the prevention or treatment of depression among 13–25-year-olds. We sought to determine in what way, in which contexts, and for whom PS appears to work in addressing depression. We found 25 studies involving 20 interventions. Results are promising for secondary prevention interventions, or interventions targeting clinical level populations, that utilize PS as the primary intervention [12, 47–49]. These studies not only found a significant reduction in depression symptoms compared to active [48, 49] and non-active [12, 47] controls but also found a significant reduction in suicidal orientation and ideation [12, 47, 49]. These findings are consistent with meta-analyses of adult PS interventions [21, 22, 23], highlighting that PS interventions for AYA can be effective in real-world settings.

For other types of interventions (i.e., universal, selective prevention, indicated prevention), results were mixed in reducing depression. The one universal program was found to have a small, significant effect in reducing depression symptoms compared to a non-active control [37]. Most selective prevention programs were not effective [39, 40, 56], and those that did show small, significant effects had mixed outcomes for follow-up maintenance [41, 42]. Most indicated prevention programs were not effective [30, 31, 45–47], yet a follow-up study showed a significant reduction in internalizing symptoms at 12-month post-treatment compared to an active control [44]. Given that these studies targeted sub-clinical populations and many of them had small sample sizes, these mixed findings may be a result of not having sufficient power to detect a meaningful difference.

Our review found limited evidence about PS skills as mediator or moderator of depression. Few studies measured improvements in PS skills; fewer still found interventions to be effective. The absence of evidence for PS abilities as a pathway is puzzling. It may be that specific aspects of PS behaviours and processes, such as problem orientation [59], are relevant. Alternatively, there may be a mechanism other than PS skills through which PS interventions influence depression.

Studies with PS as part of a wider intervention also showed mixed results, even amongst clinical populations. Although there was no clear rationale for the discrepancies in effectiveness between the studies, it is possible that the wider program dilutes the focus and impact of efficacious therapeutic elements. However, this is difficult to discern given the heterogeneity in the studies and limited information on study treatments and implementation factors. A broad conclusion might be that PS can be delivered most effectively with clinical populations in its purest PS form and may be tailored to a range of different contexts and forms, a range of populations, and to address different types of problems; however, this tailoring may reduce effectiveness.

Although the scale of impact is broadly in line with the small to moderate effectiveness of other treatments for youth depression [6], our review highlights shortcomings in study design, methods, and reporting that would allow for a better understanding of PS effectiveness and pathways. Studies varied in how well PS was operationalised. Low dosage is consistent with usage described in informal conversations with practitioners but may be insufficient for effectiveness. Fidelity was monitored in only half the studies despite evidence that monitoring implementation improves effectiveness [60]. There were references to implementation difficulties, including attrition, challenges in operationalizing online interventions, and skills of those delivering. Furthermore, most of the studies had little information about comorbidity and no analysis of whether it influenced outcomes. Therefore, we were unable to fully examine and conceptualize the ways, how and for whom PS works. More information about study populations and intervention implementation is essential to understand the potential of PS for broader dissemination.

Our review had several limitations. We excluded studies that included four treatments known to be effective in treating depression among AYA (e.g., CBT) but where the unique contribution of PS to clinical outcome could not be disentangled. Furthermore, we relied on authors' reporting to determine if PS was included: details about operationalization of PS were often scant. Little evidence addressing the fit, feasibility, or acceptability of PS interventions was found, reflecting a limited focus on implementation. We included only English-language texts: relevant studies in other languages may exist, though our post-2000 inclusion criteria may limit this potential bias due to improved translation of studies to English over the years. Finally, the heterogeneity of study populations, problem severity, comparison conditions, outcome measures, and study designs, along with a relatively small number of included studies, limits confidence in what we can say about implementation and treatment outcomes.

Overall, our review indicates that PS may have the best results when implemented its purest form as a stand-alone treatment with clinical level AYA populations; tailoring or imbedding PS into wider programming may dilute its effectiveness. Our review also points to a need for continued innovation in treatment to improve the operationalizing and testing of PS, especially when included as a part of a more comprehensive intervention. It also highlights the need for study methods that allow us to understand the specific effects of PS, and that measure the frequency, dosage, and timing of PS to understand what is effective for whom and in what contexts.

## Supporting information

**S1 File. List of excluded studies.**
(PDF)

**S2 File. PRISMA checklist.**
(PDF)

**S1 Table. Individual risk of bias assessments using cochrane RoB2 tool by domain (1–5) and overall (6).**
(PDF)

## Acknowledgments

All individuals that contributed to this paper are included as authors.

## Author Contributions

**Conceptualization:** Kristina Metz, Jane Lewis, Bryce D. McLeod, Ludvig Bjørndal, Robyn Mildon, Aron Shlonsky.

**Data curation:** Kristina Metz, Jade Mitchell, Sangita Chakraborty.

**Formal analysis:** Kristina Metz, Aron Shlonsky.

**Methodology:** Ludvig Bjørndal, Aron Shlonsky.

**Project administration:** Robyn Mildon.

**Writing – original draft:** Kristina Metz, Jane Lewis, Aron Shlonsky.

**Writing – review & editing:** Kristina Metz, Bryce D. McLeod, Robyn Mildon, Aron Shlonsky.

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
