## [Decision Letter · Decision Letter 0]

8 Mar 2023

PONE-D-23-00042Problem-solving interventions and depression among adolescents and young adults: A systematic review of the effectiveness of problem-solving interventions in preventing or treating depression among 13-25-year-oldsPLOS ONE

Dear Dr. Metz,

Thank you for submitting your manuscript to PLOS ONE. After careful consideration, we feel that it has merit but does not fully meet PLOS ONE’s publication criteria as it currently stands. Therefore, we invite you to submit a revised version of the manuscript that addresses the points raised during the review process.

We look forward to receiving your revised manuscript.

Kind regards,

Thiago Fernandes, PhD

Academic Editor

PLOS ONE

Journal Requirements:

Thank you for stating the following in the Competing Interests/Financial Disclosure * (delete as necessary) section:

"This work was supported by Wellcome Trust and was conducted independently by the evaluators (all named authors). No grant number is available. The funders had no role in study design, data collection and analysis, decision to publish or preparation of the manuscript.

" ext-link-type="uri" xlink:type="simple">https://wellcome.org/"

We note that you received funding from a commercial source: "Wellcome Trust"

Additional Editor Comments:

Please respond each comment AND highlight each of them.

Reviewers' comments:

Reviewer's Responses to Questions

**Comments to the Author**

1. Is the manuscript technically sound, and do the data support the conclusions?

Reviewer #1: Yes

Reviewer #2: Yes

2. Has the statistical analysis been performed appropriately and rigorously? 

Reviewer #1: N/A

Reviewer #2: Yes

3. Have the authors made all data underlying the findings in their manuscript fully available?

Reviewer #1: Yes

Reviewer #2: Yes

4. Is the manuscript presented in an intelligible fashion and written in standard English?

Reviewer #1: Yes

Reviewer #2: Yes

5. Review Comments to the Author

Reviewer #1: Thank you for providing a very well-written, clear, and detailed manuscript of a very interesting and worthwhile study. It was a pleasure to read and I commend you on your work. I have only a few minor comments which are mostly just points of proofreading:

- Some abbreviations in Table 1 are either not detailed before usage here, or in fact aren't expanded upon at all. Please check and detail abbreviations in your notes section (namely PST, PCT, PST-PC, NST, DSM). I recognise that readers may be familiar with some of these or could hazard a well-educated guess, but for ease of readability and for clarity it would be beneficial to amend this.

- It would be beneficial, if possible from your data, to add more detail about the ages of participant in Table 1, or to provide more details in your results section. You explained that some studies included under 13s and/or over 25s, but it is unclear which studies did so. It would also help the reader to assess the included literature more effectively if the mean/median age of participants (as per your inclusion criteria) was noted in the table.

- Lines 200-202 are unclear and confusing to read

- While I recognise the need for your review, I'm not entirely sold on your research question by your introduction section. Particularly, why you have chosen this specific intervention for this population. It may be beneficial to expand on the final paragraph (lines 83-95).

- It may be beneficial to also add your thoughts on what your results mean for clinical practitioners in your discussion. You provide some good recommendations for research, but general expansion here would be helpful.

These are the only minor edits I see as being required as your paper is strong. The results and conclusion are well written and thorough. Thank you also for providing detailed supplementary materials.

Best of luck with your ongoing work

Reviewer #2: The systematic review summarizes 25 studies concerning the efficacy of problem-solving interventions for preventing or treating depression. The topic is thoroughly examined, and the results provide insight into the development of evidence-based interventions and the enhancement of mental health outcomes for adolescents and young adults.

I have some minor concerns which I will elaborate on below:

Title:

1. The use of "13-25 years olds" in the title can be misleading as it implies a full age range rather than the mean or median age.

Introduction:

2. While the introduction is logically structured, it would be beneficial to introduce problem-solving (PS) as a technique for depression treatment early on. PS intervention is a key concept, yet it is not mentioned until the last paragraph.

3. The rationale for focusing on the effect of PS interventions on depression needs further clarification. What makes PS a more relevant technique than other techniques? I agree that maladaptive PS is associated with depressive symptoms (line 84), while the construct of PS as a coping strategy may be different from PS as an intervention technique.

4. The relationship between PS technique and evidence-based treatments is slightly confusing. EBTs such as CBT have shown small to moderate effects in preventing and treating depression (line 66), so emphasis might move to discrete treatment techniques such as PS (line 75). However, PS is usually a component of CBT, a technique used in multiple sessions. What might account for a part of the therapy being more effective than the entire therapy?

5. Line 90 refers to the complex relationship between PS and depression. Although details can be found in the results section, it would be clearer to provide a specific explanation here for “complex.”

Discussion:

6. In line 381, the authors “sought to determine in what way, in which contexts, and for whom PS appears to work in addressing depression.” However, outcomes are not discussed by context or study population. Studies conducted among students could differ from those conducted among peripartum women. Did the comparisons between contexts/populations bring forth any conclusions?

7. Among studies that found a significant reduction in depression, some reported that the effect was not sustained (e.g., line 262, line 268, line 302) while others reported the opposite (e.g., line 311, line 350). Is there a possible explanation for this discrepancy?

6. PLOS authors have the option to publish the peer review history of their article (what does this mean?). If published, this will include your full peer review and any attached files.

Reviewer #1: No

Reviewer #2: **Yes: **Tianyue Mi

---

## [Author Response · Author response to Decision Letter 0]

24 Apr 2023

See below. Also replicated in the "Response to Reviewers" document.

Thank you to the two reviewers for their thoughtful and comprehensive review of our manuscript. We have carefully considered all the comments and made requested modifications. As a result, we believe that the manuscript is improved. 

Comments to the Author

1. Is the manuscript technically sound, and do the data support the conclusions?

Reviewer #1: Yes

Reviewer #2: Yes

2. Has the statistical analysis been performed appropriately and rigorously? 

Reviewer #1: N/A

Reviewer #2: Yes

3. Have the authors made all data underlying the findings in their manuscript fully available?

Reviewer #1: Yes

Reviewer #2: Yes

4. Is the manuscript presented in an intelligible fashion and written in standard English?

Reviewer #1: Yes

Reviewer #2: Yes

5. Review Comments to the Author

Response to Reviewer 1 comments:

Reviewer #1: Thank you for providing a very well-written, clear, and detailed manuscript of a very interesting and worthwhile study. It was a pleasure to read and I commend you on your work. I have only a few minor comments which are mostly just points of proofreading:

- Some abbreviations in Table 1 are either not detailed before usage here, or in fact aren't expanded upon at all. Please check and detail abbreviations in your notes section (namely PST, PCT, PST-PC, NST, DSM). I recognise that readers may be familiar with some of these or could hazard a well-educated guess, but for ease of readability and for clarity it would be beneficial to amend this.

Thank you for your keen eye and this suggestion. We have updated the manuscript to ensure that all acronyms in the table are in the notes section.

- It would be beneficial, if possible from your data, to add more detail about the ages of participant in Table 1, or to provide more details in your results section. You explained that some studies included under 13s and/or over 25s, but it is unclear which studies did so. It would also help the reader to assess the included literature more effectively if the mean/median age of participants (as per your inclusion criteria) was noted in the table.

Thank you for this suggestion. We have added this information into the Table 1 under the “study population” information.

- Lines 200-202 are unclear and confusing to read

Thank you for this feedback. We have adjusted the sentence to hopefully increase comprehension. Please let us know if the sentence (now lines 198-199) is still unclear and/or confusing to read.

- While I recognise the need for your review, I'm not entirely sold on your research question by your introduction section. Particularly, why you have chosen this specific intervention for this population. It may be beneficial to expand on the final paragraph (lines 83-95).

Thank you for this feedback. We have re-worked the introduction section to include more information on PS and its potential as an active ingredient for AYA depression treatment. Please see lines 71-94.

- It may be beneficial to also add your thoughts on what your results mean for clinical practitioners in your discussion. You provide some good recommendations for research, but general expansion here would be helpful.

Thank you for this feedback. We have summarized the recommendations more clearly at the end of the paper. Please see lines 436-437 for clinical implications.

These are the only minor edits I see as being required as your paper is strong. The results and conclusion are well written and thorough. Thank you also for providing detailed supplementary materials.

Best of luck with your ongoing work

Thank you so much for the thorough and thoughtful review. We believe your comments aided to the creation of an improved manuscript. 

Response to Reviewer 2 comments:

Reviewer #2: The systematic review summarizes 25 studies concerning the efficacy of problem-solving interventions for preventing or treating depression. The topic is thoroughly examined, and the results provide insight into the development of evidence-based interventions and the enhancement of mental health outcomes for adolescents and young adults.

I have some minor concerns which I will elaborate on below:

Title:

1. The use of "13-25 years olds" in the title can be misleading as it implies a full age range rather than the mean or median age.

Thank you for this suggestion. We have edited the title to only include the reference to adolescents and young adults to be more fitting.

Introduction:

2. While the introduction is logically structured, it would be beneficial to introduce problem-solving (PS) as a technique for depression treatment early on. PS intervention is a key concept, yet it is not mentioned until the last paragraph.

Thank you for this feedback. We have re-worked the introduction section to have PS introduced earlier in the introduction. 

3. The rationale for focusing on the effect of PS interventions on depression needs further clarification. What makes PS a more relevant technique than other techniques? I agree that maladaptive PS is associated with depressive symptoms (line 84), while the construct of PS as a coping strategy may be different from PS as an intervention technique.

Thank you for this feedback. We have re-worked the introduction section to include more information on background treatments using PS amongst adults and its potential as an active ingredient for AYA depression treatment. Please see lines 71-94.

4. The relationship between PS technique and evidence-based treatments is slightly confusing. EBTs such as CBT have shown small to moderate effects in preventing and treating depression (line 66), so emphasis might move to discrete treatment techniques such as PS (line 75). However, PS is usually a component of CBT, a technique used in multiple sessions. What might account for a part of the therapy being more effective than the entire therapy?

Thank you for this feedback and question. We have added information to the manuscript that discusses a meta-analysis on PS within adult populations that found Problem Solving Therapy (PST) to be as effective as CBT and IPT, and more effective than WLC. We have additionally added information around the potential benefits of distilling common elements with this AYA population. Please see lines 71-94.

5. Line 90 refers to the complex relationship between PS and depression. Although details can be found in the results section, it would be clearer to provide a specific explanation here for “complex.”

Thank you for this feedback and question. Due to all the additional PS information added to the introduction, we removed this statement and only addressed in the discussion section. 

Discussion:

6. In line 381, the authors “sought to determine in what way, in which contexts, and for whom PS appears to work in addressing depression.” However, outcomes are not discussed by context or study population. Studies conducted among students could differ from those conducted among peripartum women. Did the comparisons between contexts/populations bring forth any conclusions?

Thank you for this feedback and question. Unfortunately, due to the heterogeneity in the study samples and settings as well as limited implementation factors discussed in the publications, these factors were unable to be explored. I added this limitation to lines 412-423. 

7. Among studies that found a significant reduction in depression, some reported that the effect was not sustained (e.g., line 262, line 268, line 302) while others reported the opposite (e.g., line 311, line 350). Is there a possible explanation for this discrepancy?

Thank you for this question. Unfortunately, the heterogeneity of study populations, problem severity, comparison conditions, outcome measures, and study designs, along with a relatively small number of included studies, limits confidence in what we can say about implementation and treatment outcomes. This includes an explanation for the discrepancies in sustained effects.

6. PLOS authors have the option to publish the peer review history of their article (what does this mean?). If published, this will include your full peer review and any attached files.

Do you want your identity to be public for this peer review? For information about this choice, including consent withdrawal, please see our Privacy Policy.

Reviewer #1: No

Reviewer #2: Yes: Tianyue Mi

---

## [Decision Letter · Decision Letter 1]

5 May 2023

Problem-solving interventions and depression among adolescents and young adults: A systematic review of the effectiveness of problem-solving interventions in preventing or treating depression

PONE-D-23-00042R1

Dear Dr. Metz,

We’re pleased to inform you that your manuscript has been judged scientifically suitable for publication and will be formally accepted for publication once it meets all outstanding technical requirements.

Kind regards,

Thiago P. Fernandes, PhD

Academic Editor

PLOS ONE

Additional Editor Comments (optional):

Reviewers' comments:

Reviewer's Responses to Questions

**Comments to the Author**

1. If the authors have adequately addressed your comments raised in a previous round of review and you feel that this manuscript is now acceptable for publication, you may indicate that here to bypass the “Comments to the Author” section, enter your conflict of interest statement in the “Confidential to Editor” section, and submit your "Accept" recommendation.

Reviewer #1: All comments have been addressed

Reviewer #2: All comments have been addressed

2. Is the manuscript technically sound, and do the data support the conclusions?

Reviewer #1: Yes

Reviewer #2: Yes

3. Has the statistical analysis been performed appropriately and rigorously? 

Reviewer #1: Yes

Reviewer #2: Yes

4. Have the authors made all data underlying the findings in their manuscript fully available?

Reviewer #1: Yes

Reviewer #2: Yes

5. Is the manuscript presented in an intelligible fashion and written in standard English?

Reviewer #1: Yes

Reviewer #2: Yes

6. Review Comments to the Author

Reviewer #1: Thank you for returning the manuscript with all comments addressed. I feel this manuscript is now strong and of good quality and details an interesting, well articulated, and important piece of research.

Reviewer #2: (No Response)

7. PLOS authors have the option to publish the peer review history of their article (what does this mean?). If published, this will include your full peer review and any attached files.

Reviewer #1: **Yes: **Leah Attwell

Reviewer #2: No

---

## [Editor Report · Acceptance letter]

11 May 2023

PONE-D-23-00042R1 

Problem-solving interventions and depression among adolescents and young adults: A systematic review of the effectiveness of problem-solving interventions in preventing or treating depression 

Dear Dr. Metz:

I'm pleased to inform you that your manuscript has been deemed suitable for publication in PLOS ONE. Congratulations! Your manuscript is now with our production department. 

Kind regards, 

on behalf of

Dr. Thiago P. Fernandes 

Academic Editor

PLOS ONE